# Failure Reason of PI Test Samples of Neural Implants

**DOI:** 10.3390/s23031340

**Published:** 2023-01-25

**Authors:** Jürgen Guljakow, Walter Lang

**Affiliations:** Institute for Microsensors, Actuators, Systems (IMSAS), University of Bremen, Otto-Hahn-Allee 1, 28359 Bremen, Germany

**Keywords:** neural implants, accelerated lifetime test, Polyimide

## Abstract

Samples that were meant to simulate the behavior of neural implants were put into Ringer’s solution, and the occurring damage was assessed. The samples consist of an interdigitated gold-structure and two contact pads embedded between two Polyimide layers, resulting in free-floating structures. The two parts of the interdigitated structure have no electric contacts and are submerged in the solution during the experiment. The samples were held at temperatures of 37 and 57 ${}^{\circ }$C in order to undergo an accelerated lifetime test and to compare the results. During the course of the experiment, a voltage was applied and measured over a resistance of 1 kOhm over time. Arduinos were used as measuring devices. As the intact samples are insulating, a sudden rise in voltage indicates a sample failure due to liquid leaking in between the two polyimide layers. Once a short-circuit occurred and a sample broke down, the samples were taken out of the vial and examined under a microscope. In virtually all cases, delamination was observable, with variation in the extent of the delaminated area. A comparison between measured voltages after failure and damage did not show a correlation between voltage and area affected by delamination. However, at a temperature of 37 ${}^{\circ }$C, voltage remained constant most of the time after delamination, and a pin-hole lead to a lower measured voltage and strong fluctuations. Visually, no difference in damage between the 37 and the 57 ${}^{\circ }$C samples was observed, although fluctuations of measured voltage occurred in numerous samples at a higher temperature. This difference hints at differences in the reasons for failure and thus limited applicability of accelerated lifetime tests.

## 1. Introduction

### 1.1. Historical Overview

One of the first intracortical electrode designs in use was the metal wire electrode. Its use dates back to 1950 ([1], p. 122). Several examples can be found in [2,3]. Electrodes of such type are still commercially available [4]. This type of electrode was followed by silicon-based electrodes, with the Michigan and Utah-Array as the best-known representatives of this approach ([5], p. 1387ff and [6] p. 18ff). The Utah Array is fabricated from a silicon wafer, as can be read in [7]. The in-house fabrication of a sample similar to a Michigan Array is demonstrated in [8]. An overview of the applications can be found in [9], with a concrete example in [10] that shows how it can be used to provide vision to blind people. Polyimide has been used in neural implants for decades. One of the earlier uses was as an insulant for cables, as it was patented in 1987, [11,12] or as an interconnect [13,14]. The use of PI stretched from the insulation of cables to the implants themselves, for example, cuff electrodes, as early as 1981 [15,16] or 2000 [17]. Retinal implants were another application of polyimide, such as in [18] or the Argus II implant [19]. It was used several times as a substrate for neural implants [20,21] and concrete sieve electrodes [22]. Since at least 2001, polyimide has been used for intracortical implants [23]. Several other examples, such as cuff-electrodes or epiretinal vision prothesis, are provided in [24]. Sufficient progress was made for polyimide-based electrodes to be offered commercially as ECoG implants [25,26]. As questions arose as to whether the difference in mechanical properties could lead to damage to the brain matter, new designs emerged that were fabricated from polymers ([5], p. 1381ff) [27]. The silicon probes also subsist mechanical strain, as can be seen in [28]. In the last few years, newer designs saw the light of day; among them were 3D-printed electrodes [29], Utah-array-like electrodes made of aluminum [30] and the Neuropixel implant [31], which enables the registration of more extensive amounts of data than hitherto possible due to the increased number of electrodes, now reaching close to one thousand. One design that promises to minimize the mechanical mismatch between the brain matter and the implant is the mesh-electrode, described in [32]. Other works tried to use other polymers (such as SU8 [33]), a combination of polymers and hard materials [34], having electrodes on more than one side [35], polyimide [23,36], and a range of other polymers [37,38].

### 1.2. Degradation of Neural Implants

The use of the mentioned devices leads to degradation, which necessitates an extensive analysis of those damages. One approach to gain a better understanding is the use of in-vitro methods, and the other is the in-vivo approach, whereby the probe is implanted into a model organism, e.g., rodents, felines, primates, etc., explanted, and undergoes further analyses. Combined with accelerated lifetime tests, the initial approach provides the opportunity to assess the foreseeable duration of use [39]. The latter is indispensable, as it provides information as to whether the anticipated lifetimes are achievable and which deteriorations come to pass [40]. A juxtaposition of in-vitro and in-vivo approaches was already shown for wire-electrode-probes and Utah Arrays. In [41], the analyses encompass SEM pictures and observations and how the wire electrodes affect the surrounding tissue. Similar work was undertaken on Utah Arrays [42], where the impacts of a Utah Array implant on the surrounding tissue, and vice versa, are described. First, experiences were made with samples made of PPX-C, where comparisons were made between in-vivo and in-vitro influence [40]. For PI, trials on in-vitro samples were performed, whereby the effect of temperature and other parameters on samples in a saline solution was examined [43]. The first results of in-vivo experiments are described in [44]. While there is experience with the longterm behavior of Si-based PI-insulated neural test structures [45], experience for free-floating Si-based test structures with a similar methodology is lacking. Other important points that this work focuses on and which were briefly mentioned in [45] are the failure reasons and the state of the samples after the failure.

### 1.3. This Work

As earlier described, stiff Silicon-based neural implants, e.g., Michigan-array and Utah-array, can cause problems, which lead to the development of soft implants that are made of polymers.

Apart from the mismatch in young modulus, which is several orders of magnitude higher for silicon or tungsten compared to polymers, there are several other factors that are very important for the applicability of neural implants. Next to the mismatch of a young modulus, a mismatch of density is also to be avoided [46]. Furthermore, the production of the material was found to impact the body’s reaction [47]. The shape of the neural implants is relevant, as is the bending stiffness and whether tethering is present or not [48,49]. In [50], the influence of the overall stiffness was shown as an important factor for the reaction of the brain matter. The overall stiffness includes the elastic modulus and shape of the implants. The insertion process also affects the reaction of the brain matter [46].

This tendency laid the grounds for the decision to make test structures out of polymers and omit the use of stiff silicon in this work. By using polymers, it is possible to simulate the behavior of free-floating samples. As polymers can not completely hinder any permeation of liquids, the formation of short-circuits between gold strands remains a reason for implant failure. The possible means by which a liquid can permeate are listed in [51], with an additional overview of possible effects on the electrical properties in [52].

In order to understand the failure mechanism, samples simulating the behavior of neural implants were fabricated and submerged into Ringer’s solution. The vials with the samples were heated up to 37 and 57 ${}^{\circ }$C. The assessment of damages occurring at those different temperatures allows to evaluate whether the damages are comparable and to decide if accelerated-lifetime tests are useful means in those cases.

In numerous publications, the procedure for finding out when a sample broke down was to make use of electrical impedance spectrometry (EIS) [52]. The change in the curves was interpreted as the time to failure. In this approach, the measurements take place in increasing time intervals. In this work, this approach was deemed not precise enough, and thus, an approach was chosen that is simpler to use and easier to implement and interpret. Voltage was applied to the ID structures and the time was measured till the voltage rose abruptly from 0 to above 4 V to remain permanently at that level. Once a rise in voltage was measured, the event was recorded as a failure of the sample. The samples were not taken out immediately after failure but once a month. In the work published so far, the emphasis was more on the material properties and permeation of liquids through the materials. No publication was found that described the failure of samples made from PI in a similar structure. Only the failure of the surface layer of a Utah Array, albeit with a Parylene-C (PPX-C) layer, was covered [40]. One juxtaposition of sample breakdowns in-vivo and in-vitro samples made of PI was found in [43], but nothing similar for foil-based sensors could be found.

For this work, it was assumed that pin-holes, water permeation following the polymer interface or water permeation through the polymer are the most important reasons for the intrusion of liquid between the polymer layers. Experience from previous experiments in this institute with Si-based samples hinted towards the permeation through the polymer as, in several cases, liquid pockets formed in the middle of the sample. One such example can be seen in [53], where interdigitated structures were placed in Ringer’s solution, while at the same time undergoing regular ultrasonic treatment at elevated temperatures. Another example of accelerated aging tests was performed by [54], where the samples were subjected to harsh conditions by placing them in a hydrogen peroxide solution at a temperature of 87 ${}^{\circ }$C.

Unlike those works, the samples here are free-floating, and a comparison is made between surrounding temperatures of 37 and 57 ${}^{\circ }$C. No constraints, such as H${}_{2}$O${}_{2}$ or ultrasonic treatment, are added, which allows the analysis of the influence of temperature. This limitation led to much longer testing times of up to 450 days.

The expected time for these devices remaining useful is about 20 years.

## 2. Materials and Methods

### 2.1. Sample Production

The samples consist of two polymer layers and a structured gold layer and can be seen in Figure 1 on the left side. The production process of these samples can be seen in Figure 2. In the first step, a silicon wafer was coated with polyimide (U-Varnish-S, UBE), which was cured in a vacuum at temperatures of up to 450 ${}^{\circ }$C. After curing, a 300 nm thick gold layer was deposited onto the surface via DC-Sputtering, structured (step 3), and a second layer of polyimide was deposited. The ID structure consists of 10 μm wide fingers, with 10 μm space between each. As a last step, the samples were separated by a reactive ion etch step. During the etching process, the PI layer on the contact pads was removed to make contacts possible. The interface between the two polyimide layers is depicted as a black line in step 4 in Figure 2.

In Figure 1, two similar samples can be seen that have two different numbers. The idea here was to numerate the samples in order to establish whether there is a relationship between the state of the samples and the measured current once failure occurred. This measure made analysis easier, as a large number of samples, 41, was used.

### 2.2. Sample Preparation and Experiment

The samples were lifted from the wafer, contacted with wires and electrical conductive glue, and the contacts were sealed with an epoxy-glue and partly put into vials that were later filled with Ringer’s solution so that the interdigitated (ID) part of the samples was in the solution and the part with the electric contacts was outside of the solution. The ID part can be seen in Figure 1, where it is marked with the number 2. That part was lowered into the Ringer solution so far that the ID structure was submerged. A voltage of 5 V was applied and read out by an Arduino that measured the voltage drop over a 1 kOhm resistance directly behind the sample. Figure 3 shows the circuit that was used for the measurements. A voltage of 5 V DC was chosen due to practical reasons, as earlier works [45,53] have shown, in which the damages are easier to see because the dissolution of gold takes place. The sample functions as an insulator under these conditions.

## 3. Results

Once a rise in voltage above 4 V was observed, a failure was noticed, and the samples were taken out of the vials. The voltages varied mostly from 4 to 4.9 V between the samples, rarely below 4 V. The samples were not taken out immediately but once a month. The samples were rinsed with deionized water, stayed for at least one day in deionized water, rinsed again with deionized water and isopropanol, cut out with cutting tweezers, fixed with Kapton tape on a microscope slide and covered with commercially available adhesive tape. For further analysis, an optical microscope was used. The short depth of focus of the microscope permits distinguishing two non-adhering layers, where delamination occurred.

### 3.1. State of the Samples after the Procedure

The state of the samples as prepared, which can be seen in Figure 1, is representative of all samples that were used for the experiments. On both sides, the electrodes connecting the contact pads to the interdigitated structures are clearly visible. The “fingers” of the ID structure have no direct contact. On several samples, small black spots were visible, which might be either a speck of dust or a minor defect. Defects did not lead to failure in any case of isolation or following delamination—or rather, no such observation was made. As those defects are spread evenly over the whole surface, it can be expected that resulting failures would occur at random positions over the whole surface. This was not observed. Figure 4 shows one sample where delamination took place, leading to the loss of adherence between most parts of the ID structure. In detail such damage can be seen in Figure 5. Another example is Figure 6, that shows the state of a sample after a long-term-experiment, from both sides. We see clearly that the delamination occurred on one side, closer to the edge of the sample. These two samples demonstrate the state of many samples after the experiment.

No sample was observed where comparable delamination occurred in the middle of the ID structure.

Figure 5 offers a clear example of delamination. The failure is similar to the one shown in Figure 6, only magnified. Due to the limited depth of field of the microscope, several gold strands are in focus, while the other ones are out of focus. One part of the gold structure adheres to one PI layer, while the other adheres to the other PI layer. The dislocation of the gold strands hints at a weak adhesion between gold and polymer and delamination after failure.

In 36 of 37 samples, similar delamination as in Figure 4, Figure 5 and Figure 6 was visible. There was the formation of one small liquid pocket between two fingers of the ID structure in one case. That happened at a temperature of 37 ${}^{\circ }$C, which was also the only case, when no delamination of bigger parts of the samples occurred. Apart from that case, the observed damage was always similar and independent from temperature. All samples were examined for defects beforehand. Several samples had minor defects that were deemed too small to be a reason for concern. Those defects were spread out evenly over the surface of the samples. Considering their positions, the even formation of solution-pockets over the surface of the ID structure was expected. As the delamination is either present on one side, or over the whole surface, defects or pin-holes were ruled out as a reason for failure; thus, the intrusion over the interface between the two PI layers seems the most likely. No relationship was observed between the time that the samples broke down and were taken out and the state of the samples. That means that the level of destruction of the samples could not be traced to the time that they passed in the vials after the delamination took place. Longer times did not lead to a state similar to that seen in Figure 4.

Figure 7 depicts the voltage over time of a typical failure. Typically, the voltage either increased abruptly from 0 V to above 4 V or rose over the course of several seconds from 0 V to several mV, only to rise instantly to over 4 V. Once it was over 4 V, very little changed over time and the current stayed constant or underwent minor changes, such as the slow, decelerating rise that can be seen in Figure 7. A spread of the measured voltages between 4 and 4.9 V was observed, resulting in sample resistances between 20 and 250 Ohm. Due to the scale of the time frame and employed voltage, the destruction appears as an abrupt change in voltage. In all but one case, the behavior was similar. In that exception, the failure happened between two ID-fingers, where water leaked in and formed contact between the two strands. Here, the voltage rose to a level of about 4 V and oscillated strongly. The test was stopped after slightly more than 458 days for the 37 ${}^{\circ }$C samples and 423 days for the samples that were kept at 57 ${}^{\circ }$C. The conclusion that can be drawn from these tests is that failure normally occurs in the interface between two layers, which is depicted as a black line in Figure 2. After the formation of electric contact, delamination extends sufficiently fast to lower the resistance nearly abruptly to a low level in the order of hundreds of Ohm.

### 3.2. Relation between Observed Damage and Occurring Voltage

Figure 8, Figure 9 and Figure 10 were made from samples that were held at a temperature of 37 ${}^{\circ }$C. Figure 11 and Figure 12 show samples that were kept at a temperature of 57 ${}^{\circ }$C. Despite strongly different outcomes that can be seen under a microscope, the electric resistances of the samples cannot be related to the voltage over time. There is a vast discrepancy between how voltage changes and how the samples look.

In Figure 8, two vastly different states of deterioration are depicted. While the sample on the left has visible delamination in the left part of the sample, we see only minor deterioration on the right, but with visible changes to the sample, as gold is dissolved. Over the course of several hundred hours, both samples did not have stable voltages. In both cases, the voltages changed strongly.

Figure 9 shows two samples with different outcomes concerning a similar surface. We see further that before the failure of the sample on the left, there were minor voltage surges till the voltage rose to nearly 5 V after failure. The sample on the left only has an intrusion of Ringer’s liquid, leading to a very thin film inside of the sample, while on the right, many gold strands lost adhesion to the surface. In both cases, a similar voltage was achieved.

The left side of Figure 10 shows a defect in the polyimide layer that lead to sample failure. As the failure occurred several months after the start of the experiment, it can be ruled out that the failure was present initially. On the right, we see the voltage over time. The voltage differs from the four samples above as it is only at a level close to 4 V, thus slightly lower, and it fluctuates.

Two samples with minor damages can be seen in Figure 11. While optically they look similar, their electric behavior is vastly different, as can be seen in the lower figure. The resistance of the sample on the left reaches one level and remains there with minor changes over several hundred hours. At the same time, the voltage of the sample on the left fluctuates strongly between 4 and 5 V and rarely below 4 V. While the voltage similar to sample 1 1 occurred more often in the samples that were held at a temperature of 37 ${}^{\circ }$C, sample −1 3 is more typical for samples that were held at 57 ${}^{\circ }$C. At least 8 of the 16 samples from one measurement batch show similar fluctuations, while no such fluctuation takes place at a temperature of 37 ${}^{\circ }$C, except for the sample seen in Figure 10. Considering this difference, it is realistic to assume that the results of accelerated lifetime tests are only partially transferrable between two tests at different temperatures.

The sample, seen in Figure 12, shows a sample after failure. While its state is not vastly different from the samples presented in this work, the voltage changes over time were not similar to any other observed sample. This difference might be due to a different failure reason than the other samples.

## 4. Discussion

Samples that simulate the behavior of neural implants were made and underwent a soaking test in body-tempered and heated vials filled with Ringer’s solution. After failure, the samples were analyzed under the microscope. The main reason for failure can be considered the influx of Ringer’s solution between the Polyimide layers, resulting in delamination of the layers. The layers themselves were the reasons for a sample failure only once. A promising approach would be to improve the adhesion between the two layers or to add a layer between the PI layers, which improves the adhesion. The analysis of the measured currents over time hints towards different failure reasons for samples at elevated temperatures and thus limited applicability of accelerated lifetime tests.

## 5. Conclusions

The dominant failure reason for free-floating PI test samples is the delamination of one polyimide layer from the other. Pin-holes proved to be a minor failure source, as it was only the reason for failure once out of 37 samples.

The chosen method, whereby voltage is applied and read out over a resistance, which is connected in series with the sample, proved to be well suited as it is simple to use and interpret, and a continuous measurement of a comparably big number of samples is possible.

## Figures and Tables

**Figure 1 sensors-23-01340-f001:**
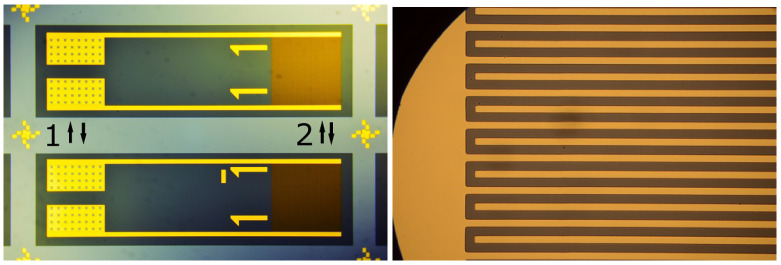
Samples that were used for the tests. (**a**) Two samples, after production; 1 marks the rectangular contact pads and 2 marks the squared interdigitated (ID) structure. (**b**) Magnified area with interdigitated structure. The structures have a width of 10 μm.

**Figure 2 sensors-23-01340-f002:**
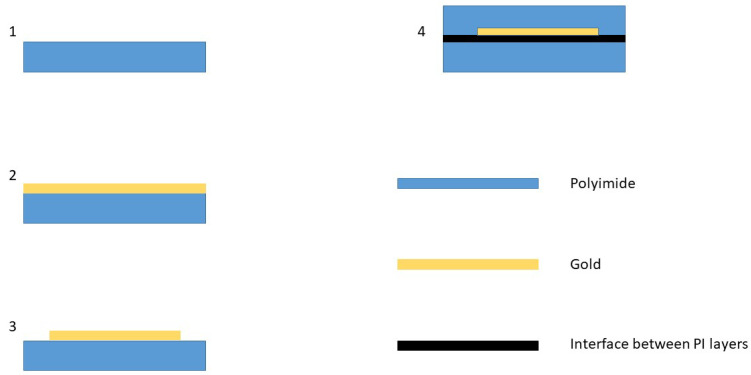
Steps in the production process of a sample. The first step is the deposition and curing of PI, followed by gold deposition (step 2), gold structuring (step 3) and spin-coating and curing of a second PI layer (step 4). Sample separation via ion etching is not depicted here.

**Figure 3 sensors-23-01340-f003:**
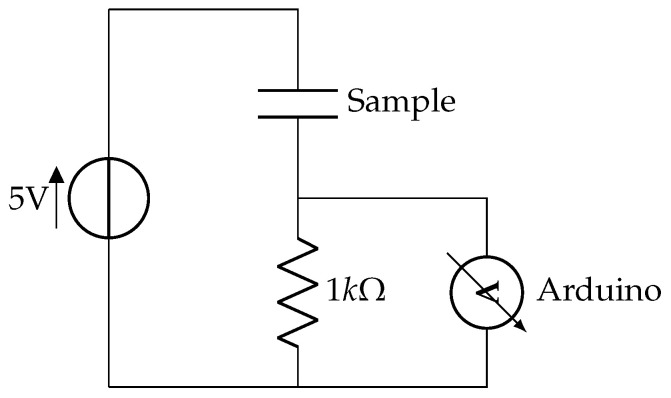
Schematic of the electric circuit used to monitor the state of the samples. A breakdown of the sample will cause a voltage step from 0 V to above 4 V.

**Figure 4 sensors-23-01340-f004:**
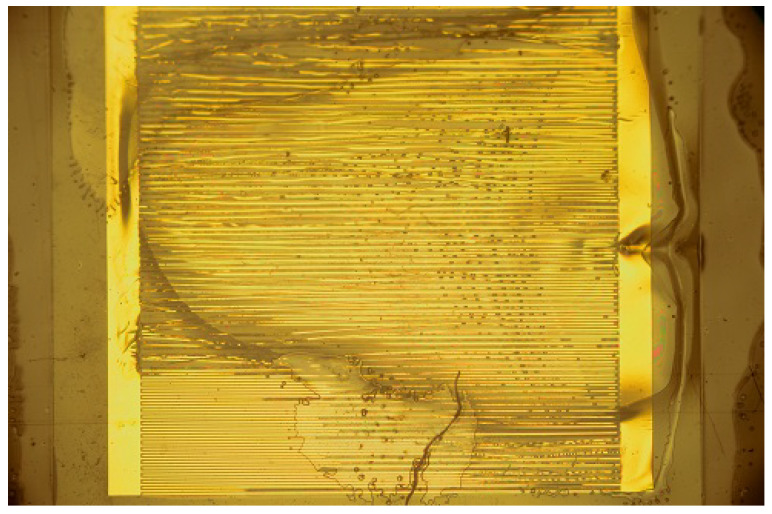
Sample after failure. Nearly the whole area used by the ID structure is delaminated.

**Figure 5 sensors-23-01340-f005:**
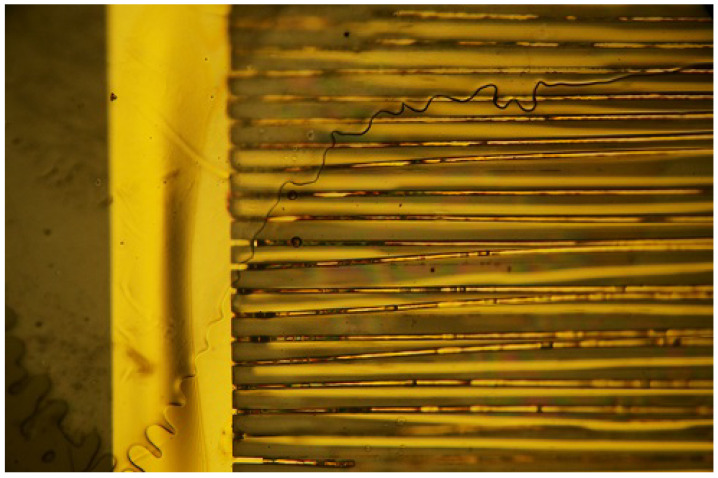
Sample after failure with visible delamination. Detachment of both layers is clearly visible as one layer is focused while the other is not.

**Figure 6 sensors-23-01340-f006:**
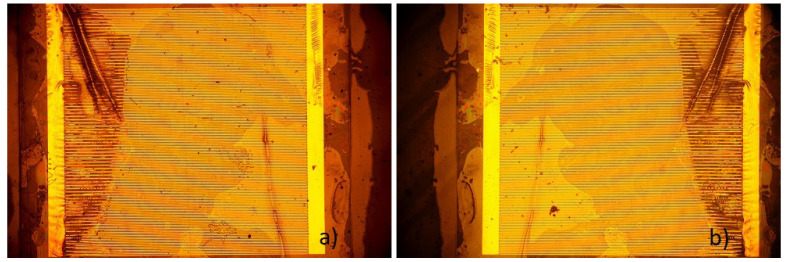
Sample after failure. Delamination is similar to most samples, as the delamination occurs on one side and leads to delamination in that area. (**a**) Sample from one side. (**b**) Same sample from the other side.

**Figure 7 sensors-23-01340-f007:**
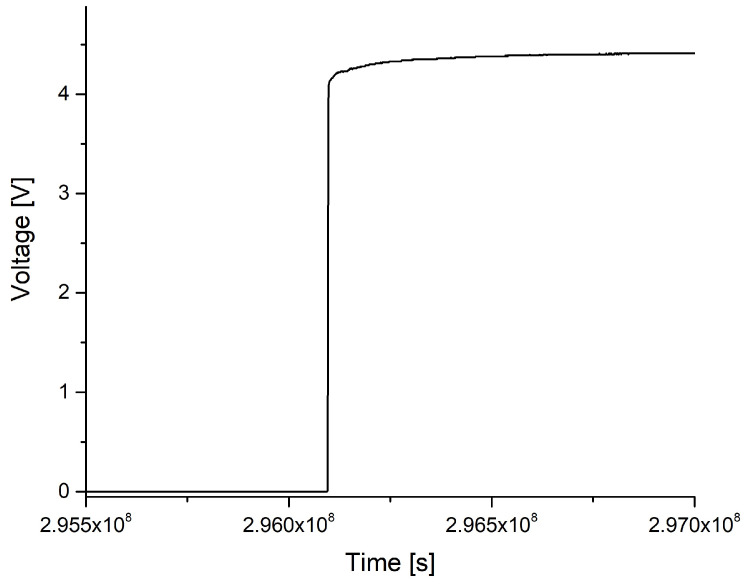
Failure over time. Measurement of voltage.

**Figure 8 sensors-23-01340-f008:**
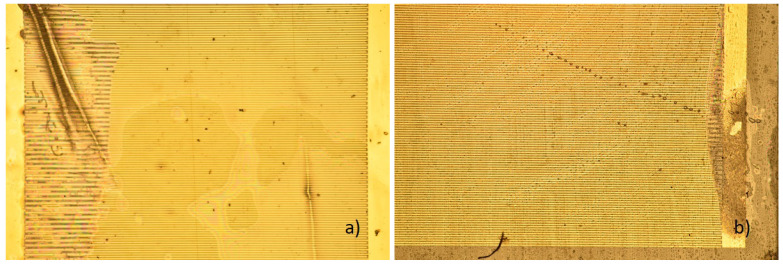
Two samples with different affected areas and strong variations in voltage. (**a**) Sample on the left, number 1 4, indicated as black. (**b**) Sample on the right, number 2 2, indicated as red (**c**) Voltage over time with visible changes.

**Figure 9 sensors-23-01340-f009:**
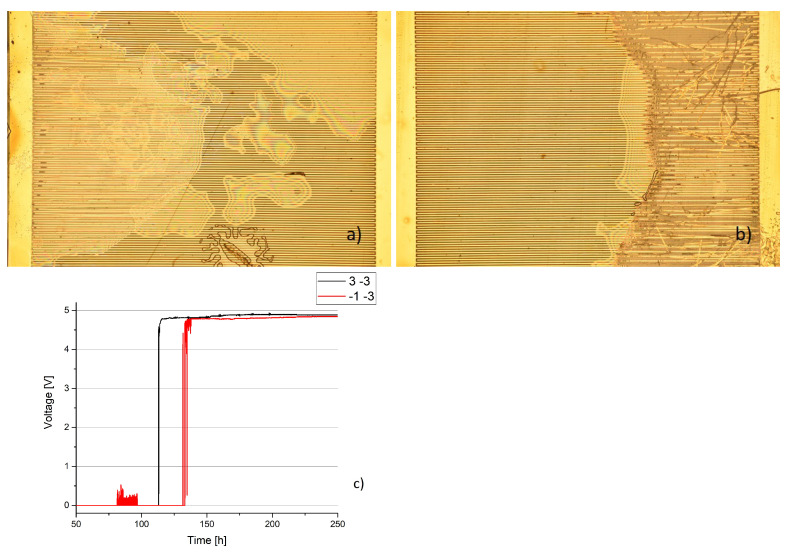
Two samples that reach a constant voltage shortly after failure. (**a**) Sample on the left, number 3 −3, indicated as black. (**b**) Sample on the right, number −1 −3, indicated as black. (**c**) Voltage over time with minor changes.

**Figure 10 sensors-23-01340-f010:**
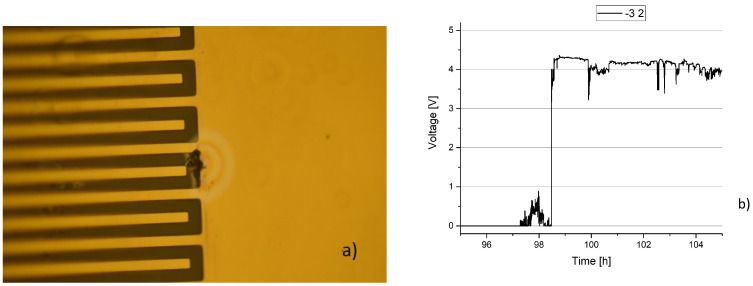
Minor defect that led to failure after 152 days. (**a**) Pin-hole in the PI layer of the sample led to failure. (**b**) Voltage over time.

**Figure 11 sensors-23-01340-f011:**
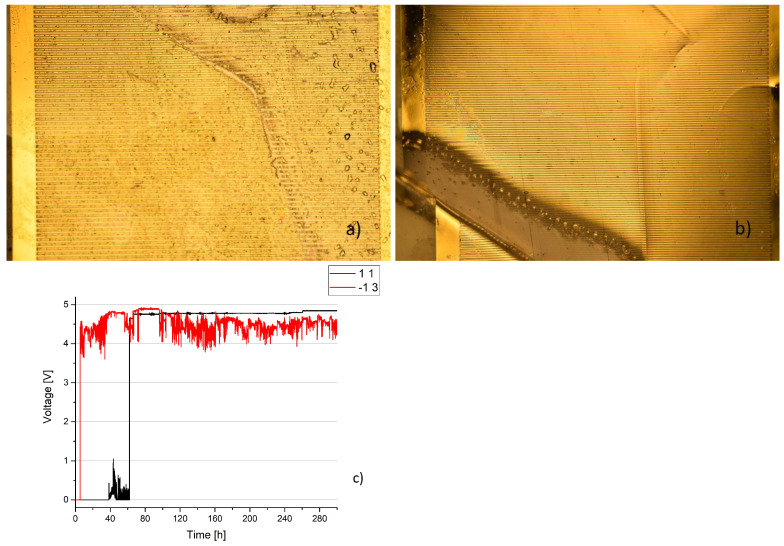
Two samples with minor deterioration. (**a**) Sample on the left, number 1 1, indicated as black. (**b**) Sample on the left, number −1 3, indicated as red. (**c**) Voltage over time, with clearly visible differences.

**Figure 12 sensors-23-01340-f012:**
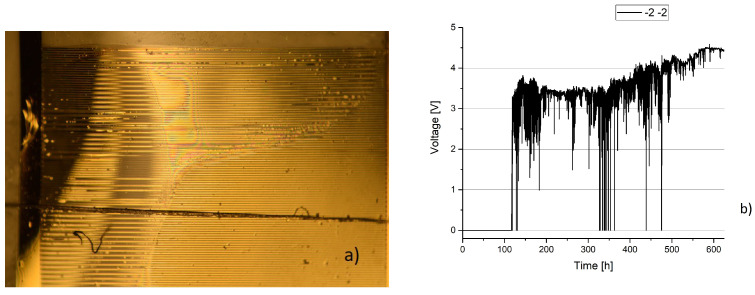
Sample 2 2. Delamination is similar to most samples, as the delamination occurs on one side and leads to delamination in that area. (**a**) Magnified area, where damage occured. (**b**) Voltage over time.

## Data Availability

Not applicable.

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
