# Peer review of "Failure Reason of PI Test Samples of Neural Implants"

_sensors, 2023, doi:10.3390/s23031340_

Round 1

Reviewer 1 Report

The submitted article addresses adhesion failure of polyimide-gold assemblies in the application case of implantable neural electrode arrays. A similar manuscript of this group has been published recently in another MDPI journal (Tintelott, M.; Schander, A.; Lang, W. Understanding Electrical Failure of Polyimide-Based Flexible Neural Implants: The Role of Thin Film Adhesion. Polymers 2022, 14, 3702. https://doi.org/10.3390/polym14183702). It should be at least cited in this manuscript and the novelty of this article compared to the previous one needs to be worked out. Content, test samples and investigations are very close. So close, that one could imagine to merge them within one publication instead of submitting two manuscripts.

The historical overview lacks the development of polyimide approaches since the story line of the submitted manuscript focuses on this material. Relevant work of several groups who have worked on polyimide or who still work on polyimide is missing, e.g. early Michigan work (J. Hetke), P. Rousche/D. Kipke, G. Loeb, T. Stieglitz Group, J. Judy Group. Retinal implants as well as ECoG and peripheral nerve electrode arrays have been developed and transferred into human clinical studies. These references and some sentences should be included to show the maturity of the state of the art before and derive the necessity of this study.

When it comes to in vitro soaking tests and accelerated ageing, P. Takmakov’s approach of using hydrogen peroxide as agent to mimic the pH shift in early foreign body reactions needs to be at least cited.

“Stiff” and “Flexible” should be more defined in detail. The state of the art is already beyond the Young’s modulus and takes bending stiffness or moment of inertia into account. Polymers are not always “soft” but might interact with less force with the target tissue. References need to be found and cited.

The authors claim to investigate whether occurring damages are comparable during accelerated ageing and body temperature. Therefore, the analysis whether the failure modes follow the anticipated Arrhenius equation with first order processes (linearity of reaction rate depending on the temperature).

The rationale to apply DC voltage of 4 V to the samples is not given. There should be a scientific hypothesis for this approach. Neural implants do not experience this amount of voltage during recording and will experience only shorter pulses during electrical stimulation application. Since the applied voltage is far beyond the water window, discussing reasons would be highly appreciated.

If pin holes are supposed to be a potential failure mode, SEM figures should be made and pinholes as well as “craters” due to water electrolysis and insulation layer destruction should be visible. Description of results and their discussion in done on a very superficial level. Distribution of failure times at the chosen temperatures is missing as well as suggestions (from literature) how to improve adhesion or how to deliver hermetic encapsulation.

Author Response

Dear Reviewer, 

Thank you for your remarks. I add as much as I could. 

Regards

Jürgen Guljakow

Reviewer 2 Report

The manuscript presents a method to investigate the failure reasons of PI interdigital electrode of neural implants. And all cases delamination of the samples were observable. The viewpoint of the article is interesting. My detailed comments on the content and suggestions to improve the quality of the manuscript are listed below.

1.       The English language of manuscript needs to be improved. Some sentences are impossible to understand.

2.       Even though you have stated that “No publication was found, that would describe the failure of samples made from PI in a similar structure.”, I believe there has to be some kind of resemblance between your findings and past similar literature/research work. It is suggested to highlight the novelty of this work in the introduction section and add discussion part.

3.       As shown in Fig.1, why two samples in left picture, aren't they the same? And it will be better to add text annotation.

4.       As shown in Fig.2, the black line should be marked with legend.

5.       Since the delamination is the important failure reason of PI-gold sensor, the failure area should be detail discussed. In particular, the influence of different failure areas on the measured voltage.

6.       The words describing illustration in the paper should be revised from picture to figure. And please revised the Fig. numbers in the text correspond to the illustrations.

7.       The results seem to lack data support; it is suggested to quantify more data. For example, in Figure 7, multiple sets of data should be added, and error bars added.

8.       The authors stated that “The result of this experiment makes it possible, to direct future efforts for improvements of neural implants.”, however, the experiment for neural implants or its association with the developed IP-gold sensor I found missing in your Results part.

Author Response

Dear Reviewer, 

Thank you for your remarks. 

Regards

Jürgen Guljakow

Reviewer 3 Report

" Failure reason of PI test samples of neural implants”, Article reference: sensors-2015268

 This study discusses damage and measurement of neural implants placed in Ringer-solution However, the authors do not address many of the physiological needs and conditions associated with the environment of neural implants, nor discuss post-injury discussions. The paper quality is not suitable for publication in this journal

 REFEREE REPORT:

 1.     Please provide environmental requirements and physiological conditions for neural implants.

2.     Why did the author use Ringer-solution as the test solution? Are there any relevant regulations and references mentioned?

3.     References that do not mention relevant accelerated aging experiments.

4.     The electrode of the neural implant is exposed to the tissue fluid, which will cause delamination and affect the lifespan, which is not mentioned in this study.

5.     What is the expected lifespan of the neural implant? The study did not introduce how much lifespan the design would have in actual use.

6.     This study does not discuss the equivalent circuit model, nor does it discuss the behavior of solution penetration in the equivalent circuit when performing the soaking test.

Author Response

Dear Reviewer, 

Thank you for your remarks. I added several paragraphs that I hope will answer the points, that you remarked. 

regards

Jürgen Guljakow

Round 2

Reviewer 2 Report

The authors have addressed the comments in the previous review report, so I suggest it is acceptable by now.

Author Response

Dear Reviewer, 

Thank you for your review. 

Regards

Jürgen Guljakow

Reviewer 3 Report

1.     The writing structure of the abstract in this study is incomplete, and the experimental results are not clearly mentioned. It is recommended to modify the abstract.

2.     It is unacceptable in the writing of the paper that this research only discusses without conclusion.

3.     Reference 11 is the same as 36.

Author Response

(The authors gave the same response as above.)

Round 3

Reviewer 3 Report

In writing journal papers, it is almost impossible to see the writing of conclusions using the column method, which means that the author has little experience writing journal papers. It is recommended that the authors refer to the writing of other papers in this journal (Sensors) for correction.

Author Response

Dear Reviewer, 

Thank you for your reply. 

The conclusions-part was changed from enumeration to test-form. As I have seen one conclusion written as an enumeration I thought, that was possible.  

Regards

Jürgen Guljakow